# Examining the Effectiveness of the Discharge Plan Model on the South Korean Patients with Cancer Completed Cancer Treatment and Are Returning to the Community: A Pilot Study

**DOI:** 10.3390/ijerph20010074

**Published:** 2022-12-21

**Authors:** Young Ae Kim, Hye Ri Choi, Mingee Choi, Ah Kyung Park, Hye Ryun Kim, Chaemin Lee, Elim Lee, Kyung Ok Kim, Mi Young Kwak, Yoon Jung Chang, So-Youn Jung

**Affiliations:** 1National Cancer Control Institute, National Cancer Center, Goyang-si 10408, Republic of Korea; 2School of Nursing, University of Hong Kong, Hong Kong; 3Graduate School of Social Welfare, Yonsei University, Seoul 03722, Republic of Korea; 4Department of Social Work, National Cancer Center, Goyang-si 10408, Republic of Korea; 5Department of Social Welfare, Seoul Welfare Foundation, Seoul 04147, Republic of Korea; 6College of Nursing, Kyungbok University, Namyangju-si 12051, Republic of Korea; 7Public Healthcare Quality Improvement Team, National Medical Center, Seoul 04564, Republic of Korea; 8Center for Breast Cancer, National Cancer Center, Goyang-si 10408, Republic of Korea

**Keywords:** neoplasms, survey, discharge, questionnaire, health service access and utilization

## Abstract

This study aimed to examine the effectiveness of a discharge plan model for South Korean patients with cancer who had completed treatment and were returning to the community. Overall, 23 patients with cancer were recruited at the National Cancer Center in Goyang-si. The effectiveness of the discharge plan was examined using four methods: Social Needs Screening Toolkit (2018), early screening for discharge plan, current life situation v.2.0, and a questionnaire regarding problems after discharge from the hospital. Subsequently, the results were analyzed using descriptive statistical analysis methods with the Stata 14.0 program. The largest age group of study participants was between 45 and 64 years. No participants responded to urgent needs, whereas nine (39.13%) participants needed support for their social needs. According to the in-depth evaluation of participants, more than 80% of the respondents answered that patients with cancer needed no help in self-management, daily living activities, or mental health. The satisfaction survey results showed that the degree to which the “discharge plan” was helpful for health management at home after discharge was 4.41 of 5, and the degree to which it helped return to daily life was 3.86 of 5.

## 1. Introduction

The number of patients newly diagnosed with cancer has increased annually in South Korea following increased early screening [1,2]. Although the new cancer diagnosis in 2017 was 235,547 cases, the number of cases increased from 3.5% to 243,837 in 2018 [3]. Similarly, the 5-year relative survival rate of cancer has increased by 54.1% between 2001 and 2005 and 70.3 between 2014 and 2018 [3]. However, since the life of cancer survivors is extended, they have to face various challenges in their lives after cancer treatment, such as physical, psychological, and emotional issues, and return to work [4,5,6,7]. The challenges are further deteriorated by poor accessibility if the residences of cancer survivors are remote from care centers. [8] Therefore, care plans for cancer survivors were proposed by sharing the needs for integrated care in the transition from cancer treatment to primary medicine and the community to support them after treatment [9,10,11].

In South Korea, community-based health and welfare integrated support was established as a part of the “Inclusive Korea” policy implemented in 2018 [12]. Accordingly, the Ministry of Health and Welfare has implemented policies to resolve regional disparities by providing and linking essential treatment and care, including categorizing the country into regions and designating medical centers in charge of each region. One such support is the discharge plan, which provides linking services of health and welfare in the community after the discharge of patients with stroke, cardiopathy, cancer, geriatric fractures, and respiratory diseases [13]. The discharge plan is currently in the early stage, promoted by the Ministry of Health and Welfare and the Health Insurance Review and Assessment Service, and was implemented in 2020 by providing information on community health and welfare resources to patients with cerebrovascular diseases.

The discharge plan is an essential service for patients with cancer returning home after cancer treatment because many patients fail to manage their health after discharge [14]. Therefore, they face risks of complications, recurrence, metastasis, secondary cancer, and psychosocial problems, including emotional anxiety, depression, suicidal thoughts, and distress [14,15,16]. Notably, patients with unmet social needs and many social risk factors may experience severe physical, emotional, and social difficulties after discharge [17,18]. Unmet social needs and high social risks can result in unnecessary emergency department visits and unplanned hospitalization [19]. This vicious circle can be broken by establishing a discharge plan for the patient in advance, which can screen and identify the patient with cancer’s social needs and risk factors to link community resources and develop potentially necessary resources.

Therefore, this study aimed to develop a “discharge plan” model for establishing a community-tailored treatment plan for patients with cancer and a patient evaluation tool for determining the range of community-based services needed. Furthermore, this study can provide a reasonable basis for relevant policies in the future by identifying the ranges and processes of the service, the roles of dedicated professionals, and potential limitations. To this end, this study recruited 23 patients with cancer at the National Cancer Center.

## 2. Materials and Methods

This study presents a research design that examines the effectiveness of a discharge plan model in domestic patients who have completed cancer treatment and are returning to the community. This study recruited 23 patients with cancer, collected their diagnosis, pathology, staging, history of surgery and treatment, and subsequently provided 90 min of instruction about the discharge plans and information by the research team, which included a primary investigator, nurse, office manager, medical assistant manager, and medical assistant. This study examined a service model where a multidisciplinary team composed of physicians, nurses, and social workers provides and monitors essential medical, social, and community-based information to patients based on their evaluation from admission to discharge.

The overall process of the “discharge plan” is shown in Figure 1. A physician in charge of patients with cancer asked the patients about their willingness to participate in the “discharge plan” when the patient’s hospitalization was decided. A nurse who was in charge of the “discharge plan” obtained consent from the patients with cancer who agreed to participate, and the “discharge plan” team conducted an initial screening on the day of the hospitalization decision to evaluate the patient’s basic physical condition via the early screen for discharge planning (ESDP) and the social status of the patient. On the third day after the surgery of the patient with cancer, the “discharge plan” team evaluated the patient’s physical, medical, and nursing needs again and assessed the specific social status and unmet needs. Subsequently, the integrated “discharge plan” was established at a multidisciplinary meeting to provide information on available medical and social resources and linkages. After the patient with cancer was discharged, the progress of the integrated “discharge plan” was monitored three times, 1 day after discharge, the first outpatient visit, and 1 month after discharge, to screen the updated medical and psychosocial issues and to provide a treatment plan and information.

The inclusion and exclusion criteria for the “discharge plan” study areas below

Inclusion criteria⸰Patients who were hospitalized for scheduled surgery aiming for treatment after the diagnosis of cancer (colorectal, gynecologic, gastric, breast, and lung cancers)⸰Patients who were 19 years of age or older⸰Patients who could communicate, understand the purpose and contents of the study, and voluntarily consent to participate in the study in writing⸰Patients who visited the National Cancer Center 1 month after discharge for outpatient visitsExclusion criteria (If any of the following, excluded)⸰Patients with metastatic or terminal cancer⸰Patients who were scheduled for palliative surgery due to the stage 4 cancer⸰Patients who planned another hospitalization after discharge⸰Patients who refused to participate in this study⸰Patients who could not communicate with researchers

The “discharge plan” team consisted of a “discharge plan” physician, a “discharge plan” nurse, and a “discharge plan” social worker. The physician played the role of a supervisor in establishing an integrated discharge plan by identifying the health status and medical needs of the patient with cancer who participated in the study. The duration of the intervention per patient was approximately 70 min. The “discharge plan” nurse was given approximately 280 min per patient with cancer who participated in this study to evaluate their health status and medical and nursing needs, to provide and monitor post-treatment care and manage information, such as wound care, diet, nutrition, and workout. Finally, the “discharge plan” social worker spent approximately 190 min per patient with cancer who participated in this study to provide and link information on social welfare services that can be used after discharge considering their social and economic status.

The “discharge plan” was provided to the patients with cancer who participated in cooperation with the patient’s physician, who shared the treatment plan with the team and the patients and informed them about the “discharge plan” process. In addition, the “discharge plan” team may request cooperation from additional medical staff, such as psychiatrists, physiatrists, ward nurses, primary clinic physicians, psychologists, nutritionists, rehabilitation counselors, therapists, multidisciplinary teams, and visiting nurse teams if necessary, to provide a service in need of the patient.

The four tools referenced to evaluate a patient’s status in the “discharge plan” process are as follows:(1)Initial evaluation
-Social Needs Screening Toolkit [20]

The international growth of health disparity according to social determinants is a globally noticed topic that also significantly impacts South Korea. The “5th Comprehensive National Health Promotion Plan”, announced by the Ministry of Health and Welfare on 27 January 2021, presents a vision and goal to reduce health disparities and enhance health equity by identifying and intervening in the social determinants of health. Although various studies are being conducted on social determinants of health, the social determinants are very difficult to standardize because socioeconomic, cultural, and environmental conditions differ by community and environment. In this study, we attempted to evaluate social determinants in the context of South Korean patients with cancer by referring to the Social Needs Screening Toolkit developed earlier for screening social needs by Health Leads Inc. The Social Needs Screening Toolkit is an evaluation tool published in 2016 that was developed and utilized for 20 years of experience and research. This screening toolkit categorizes essential social needs domains into seven domains: food insecurity, housing insecurity, utility needs, financial resource strain, transportation challenges, exposure to violence, and sociodemographic information domains following optional needs domains, including childcare, education, employment, health behavior, social isolation and support, and behavioral/mental health.

This study’s initial social needs screening tool consisted of 10 items developed based on the Social Needs Screening Toolkit translation. This study employed the domains of financial resource strain, “social isolation”, “housing instability”, “food insecurity”, “transportation challenges”, “utility needs”, “employment”, “childcare”, “understanding of information”, and “emergency support”. Each question was answered with either yes or no. In addition, for those who answered that they needed emergency support, the question of whether they would like to receive support, which could be answered as either yes or no, was added.

(2)In-depth evaluation of social needs: your current life situation (YCLS) [21]

This study’s in-depth evaluation of social needs was categorized into a basic social history survey and a social risk factor evaluation. The social history survey questions included basic personal information about the patient’s residence, housing type, marital status, disability, and insurance type. The social risk factor assessment questionnaire was developed based on the YCLS survey produced by Kaiser Permanente, which is a tool developed in 2016 to examine the social needs of populations at risk of high healthcare utilization. The YCLS classified social needs into 32 items under six domains so that primary medical staff could identify the patient’s social needs. Therefore, this study produced an evaluation questionnaire with the following 10 domains: economic stability, social isolation, mental health, housing, goods, transportation, understanding information, employment, social network, and care responsibility. In addition, medical centers, community referrals, community resources, and information were provided to patients with cancer who participated with more than one social risk factor.

-Detailed evaluation in the nursing area

An evaluation tool was developed based on the Problems After Discharge Questionnaire (PADQ) tool [22] to gain an in-depth understanding of the unmet nursing needs of the patients with cancer who participated for physical, psychological, informational, and socioeconomic needs. The original copyright of the PADQ-E tool was acquired for modification, and evaluation items were classified based on the results of in-depth interviews of patients with cancer who were discharged after cancer surgery at the National Cancer Center. This evaluation form included physical symptoms, self-management, daily living activities, and informational and psychological domains, indicating the extent to which assistance was needed due to unresolved difficulties while staying at home after discharge.

### Data Analysis

This study began with patients with 25 cancer from the National Cancer Center who consented to participate. However, the final number of participants who completed the study and were included in the analysis was 23 because two patients dropped out. This study presented the demographic characteristics of the participants, their social and nursing needs, the evaluation results from the medical staff, and the satisfaction survey results using descriptive statistical analysis methods. The collected data were analyzed by the descriptive statistical method using Stata 14.0.

## 3. Results

### 3.1. Demographics of Study Participants (Table 1)

The gender of the study participants was 13 (56.52%) females, which was higher than that of males, and the age group of 45–64 years (52.17%) was the largest, with 12 (52.17%). The planned goal of enrolling was five patients for each of the five different cancer types; however, two patients dropped out. Therefore, only four patients with cancer who participated were enrolled in cervical and lung cancers. Regarding the participants’ residence, 15 (65.21%) patients resided near the hospital, and the National Health Insurance Service covered 21 (91.30%). The household income of nine respondents (40.91%) was 200% lower than the national median income, accounting for the largest proportion.

### 3.2. Initial Health Conditions Evaluation: ESDP

To evaluate the initial health conditions of patients with cancer who participated in this study, an early screening for discharge plan (ESDP) was adopted, which employed a scoring system based on age, disability, living alone, and self-rated walking limitation (range: 0–23). Therefore, the overall health conditions of patients with cancer who participated were evaluated (Table 2).

### 3.3. Overall Evaluation Results

Because of the evaluation of the basic social needs of the 23 study participants, the need for care for other family members was the highest with four (17.39%), followed by utility bill- and employment-related issues with three (13.04%). In contrast, no participants responded to urgent needs, nine (39.13%) needed support for the social needs mentioned above. The average of overall evaluation result about the “discharge plan” from the study participants was 5.04, the minimum maximum values were 0 and 12, respectively.

### 3.4. In-Depth Evaluation Results (Table 3)

Because of the in-depth evaluation of the study participants, regarding the social needs of patients evaluated by social workers, care responsibility was the highest with five (21.74%), followed by financial problems with four (17.39%). According to the nurses’ in-depth evaluations, more than 80% of the responses indicated that patients with cancer did not need help with self-management, daily living activities, and mental health. However, 50% answered that patients with cancer needed partial help following possible physical activity, symptoms requiring hospital visits, contact information for inquiries, disease status and treatment plan, food, hospital visits, and examination schedules in the informational domain.

### 3.5. Satisfaction Survey Results (Table 4)

The result of the overall satisfaction survey on study participants 1 month after discharge was an average of 4.32 points of 5. The degree to which the “discharge plan” was helpful for health management at home after discharge was 4.41 of 5, and the degree to which it helped return to daily life was 3.86 of 5. Regarding detailed opinions on this, nine (39.13%) reported, “it was good to be able to ask questions that I could not ask during the treatment time”, and three stated, “I felt emotionally stable because of someone who took care of me” people (13.04%).

## 4. Discussion

This study established a “discharge plan” process in the South Korean context by referring to relevant international studies and developed an overall and in-depth evaluation form for each process. According to this process, a multidisciplinary patient support team composed of physicians, nurses, and social workers identifies the services needed during hospitalization and establishes a person-centered discharge plan for patients with cancer who have completed cancer surgery and treatment. The “discharge plan” aimed to improve the patients with cancer quality of life and to increase their confidence and knowledge about management after discharge. Accordingly, the study was designed to evaluate the unmet needs of discharged patients with cancer to verify the effectiveness of the “discharge plan”.

The “discharge plan” is an essential service for patients with cancer returning home after discharge following its contributions to preventing and resolving the patient’s physical, emotional, and social challenges after discharge.

This study’s results showed that most patients with cancer lived near hospitals (65.21%) and were covered by the National Health Insurance Service (91.30%). Regarding the patients with cancer who participated for social needs in the overall evaluation, the need for caring for family members (17.39%) was the highest, one (4.35%) had financial resource strain, and three (13.04%) had problems with utility bills. However, from the in-depth social needs evaluation by the social worker, four (17.39%) had financial resource strain, and no participants had problems with utility bills. Therefore, the gap between the overall evaluation of social needs by patients with cancer who participated and the in-depth assessment of social needs by the social worker should be studied to improve the “discharge plan” evaluation in South Korea.

The results of the nursing needs at the initial evaluation of the “discharge plan” were an average of 5.04, and the general health condition of most patients was good. Although a previous study stated that discharge planning is required for patients with a score of over 10, only two patients with cancer scored over 10 in this study. Since most study participants were in good health condition, more than 80% of the participants answered that they did not have nursing needs for health-related self-management, daily living activities, and psychology domains but had partial informational needs.

Because of surveying the satisfaction of the patients with cancer 1 month after discharge, the average overall satisfaction was 4.32 of 5, and the degree to which they assumed it was helpful in health management was 4.41 of 5. However, the degree to which they believed it helped them return to daily life was as low as 3.86 of 5. Therefore, it appears necessary to examine the reasons for this gap in satisfaction through in-depth interviews in the future.

## 5. Conclusions

To verify the effectiveness of the “discharge plan”, this study evaluated the unmet needs of patients with cancer after hospitalization. Although this study had a limitation in generalizability due to the small number of populations, it provided a foundation for a “discharge plan” in the South Korean context. Therefore, South Korean hospitals providing treatment and care to patients with cancer can be inspired by this study to consider the needs of these patients after hospitalization. However, future studies are required to validate the ‘discharge plan’ in South Korea with a larger population with various types of cancer.

## Figures and Tables

**Figure 1 ijerph-20-00074-f001:**
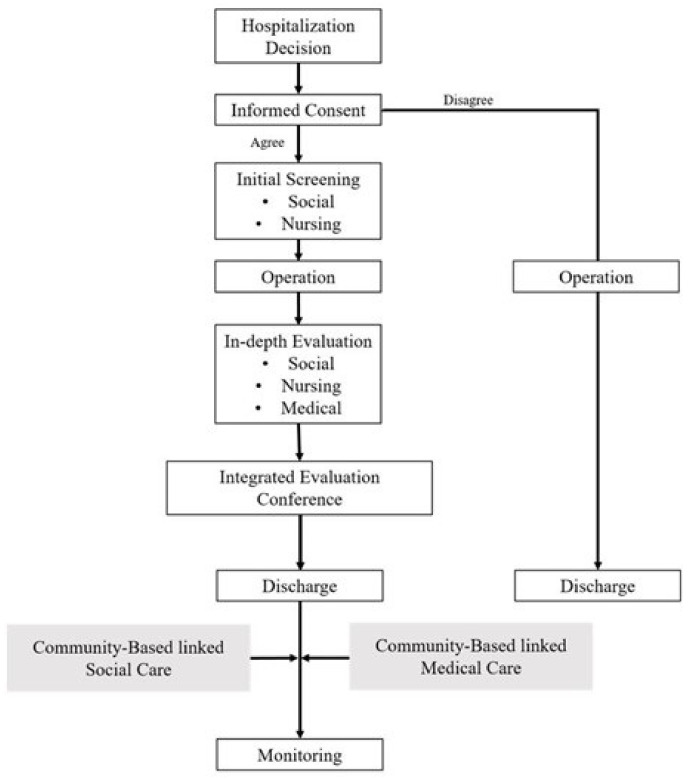
The overall process of the “discharge plan”.

**Table 1 ijerph-20-00074-t001:** Demographics of study participants.

		N	%
**Gender**	Male	10	43.48
	Female	13	56.52
**Age**	19–44	3	13.04
	45–64	12	52.17
	65–74	6	26.09
	75-	2	8.70
**Type of Cancer**	Colorectal cancer	5	21.74
	Gynecologic cancer	4	17.39
	Gastric cancer	5	21.74
	Breast cancer	5	21.74
	Lung cancer	4	17.39
**Region**	Hospital nearby (Goyang, Gyeonggi-do Province)	15	65.21
	Long distance from the hospital	8	34.78
**Type of Insurance**	Mandatory Social Health Insurance	**21**	**91.30**
	Medical Aid	2	8.70
**Household Income**	Average House Income (Mean ± Sd)	483.52 ± 330.17	
	Median Income * below 40%	5	23.81
	Median Income * below 50%	0	0.00
	Median Income * below 75%	1	4.76
	Median Income * below 100%	1	4.76
	Median Income * below 200%	**9**	**40.91**
	Median Income * 200% or more	6	26.09
Total		**23**	**100**

* Median income: Average income among the economic income population in Korea; median income is equivalent to a median income of 100%.

**Table 2 ijerph-20-00074-t002:** Initial Screening Results.

			N	%
**Basic** **social** **needs** **screening**	**Yes/No**	In the past 12 months, have you had any difficulty getting medical attention due to the cost?	1	4.35
		Do you frequently feel like you are not close to anyone?	1	4.35
		Are you worried that there will be a stable place to live?	2	8.70
		In the last 12 months, did you ever eat less than you felt you should because there was not sufficient money for food?	0	0.00
		In the past 12 months, have you not been able to see a doctor because there was no way to come and go?	0	0.00
		Need help reading hospital materials (such as brochures, doctors or pharmacists)?	1	4.35
		In the past 12 months, have you ever had difficulty using electricity, gas, water and telephones because you have not paid your bills?	3	13.04
		Do you need help finding a job or keeping a job?	3	13.04
		Are you looking after children under the age of 19 or people with chronic diseases or physical or mental disabilities?	4	17.39
		Are any of your needs urgent?	0	0.00
		Do you want to get support for these needs among the checks above?	9	39.13
Early Screen for Discharge Planning (Range: 0–23)		(Mean ± Sd)	5.04 ± 2.82 Min: 0 Max: 12	

**Table 3 ijerph-20-00074-t003:** In-depth Evaluation Results.

					N	%
In-depth assessment of social needs [Kaiser Permanente, Your Current Life Situation (YCLS) v 2.0]	Patient needs evaluated by social workers (Yes/no)	Financial			4	17.39
		Social isolation			2	8.70
		Residence			0	0.00
		Meal			1	4.35
		Transportation			1	4.35
		Understanding information			0	0.00
		Bills of utility (electricity, gas, water, and telephones)			0	0.00
		Job			1	4.35
		Responsibility for taking care of your family			5	21.74
Questionnaire Regarding Problems after Discharge from the Hospital (PADQ-E)			No help needed N (%)	Need partial assistance N (%)	Need full help N (%)	
	Physical symptoms	Pain	20 (86.96)	2 (8.70)	1 (4.35)	
		Tired	21 (91.30)	2 (8.70)	0 (0.00)	
		Discomfort in breathing	21 (91.30)	2 (8.70)	0 (0.00)	
		Bowel problems	18 (78.26)	4 (17.39)	1 (4.35)	
		Urination problems	20 (86.96)	3 (13.04)	0 (0.00)	
		Lacking appetite and indigestion	20 (86.96)	3 (13.04)	0 (0.00)	
		Hand and foot numbness and aching	21 (91.30)	2 (8.70)	0 (0.00)	
		Vomiting	22 (95.65)	1 (4.35)	0 (0.00)	
		Change in appearance	23 (100.00)	0 (0.00)	0 (0.00)	
	Self-management	Use of crutches, pedestrians, and wheelchairs, among others	23 (100.00)	0 (0.00)	0 (0.00)	
		Use of medical equipment	23 (100.00)	0 (0.00)	0 (0.00)	
		Management of stent	23 (100.00)	0 (0.00)	0 (0.00)	
		Management of Piping Pipes	22 (95.65)	1 (4.35)	0 (0.00)	
		Wound management	20 (86.96)	3 (13.04)	0 (0.00)	
		Medication Instruction	22 (95.65)	1 (4.35)	0 (0.00)	
		Exercise and Rehabilitation Instructions	20 (86.96)	3 (13.04)	0 (0.00)	
		Dietary management	18 (78.26)	5 (21.74)	0 (0.00)	
		Management of side effects of treatment (anti-cancer, radiation, surgery)	18 (78.26)	4 (17.39)	1 (4.35)	
		Management of comorbidity	20 (86.96)	2 (8.70)	1 (4.35)	
	Daily activities	Get dressed and take off	23 (100.00)	0 (0.00)	0 (0.00)	
		Shower or bath	22 (95.65)	1 (4.35)	0 (0.00)	
		food intake	22 (95.65)	1 (4.35)	0 (0.00)	
		Meal preparation	18 (78.26)	4 (17.39)	1 (4.35)	
		Lying or getting up on the bed	23 (100.00)	0 (0.00)	0 (0.00)	
		Toilet movement and use	23 (100.00)	0 (0.00)	0 (0.00)	
		Taking a walk	22 (95.65)	1 (4.35)	0 (0.00)	
	Informative	Information about the possible degree of physical activity	9 (39.13)	14 (60.87)	0 (0.00)	
		Ways to Control Pain	12 (52.17)	11 (47.83)	0 (0.00)	
		Symptoms requiring hospital visits	8 (34.78)	18 (65.22)	0 (0.00)	
		Contacting information of the hospital	9 (39.13)	14 (60.87)	0 (0.00)	
		Disease condition and treatment plan	9 (39.13)	14 (60.87)	0 (0.00)	
		How to take medication, side effects	12 (52.17)	11 (47.83)	0 (0.00)	
		food to eat and food to avoid	9 (39.13)	14 (60.87)	0 (0.00)	
		Hospital visiting schedule	11 (47.83)	12 (52.17)	0 (0.00)	
	Mental	Insomnia	19 (82.61)	4 (17.39)	0 (0.00)	
		Depression	21 (91.30)	2 (8.70)	0 (0.00)	
		Anxiety	21 (91.30)	2 (8.70)	0 (0.00)	
		Degradation of memory and thinking skills	23 (100.00)	0 (0.00)	0 (0.00)	

**Table 4 ijerph-20-00074-t004:** Patient Satisfaction Survey Results (*n* = 22, missing = 1).

How satisfied are you overall with your participation in the “discharge plan for patients with cancer?” 0 (Not satisfied)–5 (Very satisfied)	(Mean ± Sd)	4.32 ± 0.65
Do you think the “discharge plan for patients with cancer helped you manage your health at home after discharge?” 0 (Not satisfied)–5 (Very satisfied)	(Mean ± Sd)	4.41 ± 0.67
Do you think the “discharge plan for patients with cancer helped you return to your daily life after discharge?” 0 (Not satisfied)–5 (Very satisfied)	(Mean ± Sd)	3.86 ± 0.94

## Data Availability

The raw data supporting the conclusions of this article will be made available by the authors without undue reservation.

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
