# Peer review of "Examining the Effectiveness of the Discharge Plan Model on the South Korean Patients with Cancer Completed Cancer Treatment and Are Returning to the Community: A Pilot Study"

_ijerph, 2022, doi:10.3390/ijerph20010074_

Round 1

Reviewer 1 Report

I would like to thank the editor for offering this opportunity to review this manuscript. The purpose of this study is “to examine the effectiveness of the discharge plan model on South Korean cancer patients who have completed cancer treatment and are returning to the community. This is an interesting and important topic. The methodology is robust but several unclear areas require clarification and further discussion before it is publishable.

 1.     My main concern is that the study findings cannot support the study conclusion or examine the effectiveness of the discharge plan model. As a matter fact, the authors did not draw conclusion, just simply present the survey findings. As the authors stated under the conclusions section, “This study has some limitations in study participants who were recruited at a single hospital, the National Cancer Center, limited types of cancers to colorectal cancer, gynecologic cancer, gastric cancer, breast cancer, lung cancer, and the small participant number of 23.” In addition, “Since the predominant number of study participants were in good health condition…” which also limits the generalizability of the study findings.

2.     Ref is needed to support the related description of the middle paragraph (lines 58-68) on page 2.

3.     There existed contradiction regarding the recruited patents numbers on page 2 (line 74 and line 78), please clarify. In addition, lack of power analysis or sample calculation.

4.     Ref is needed for the study evaluation measures, if they are not the original developed measures of the present study.

Author Response

Dear reviewer

Thank you very much for your valuable comments. All authors endeavoured to improve the manuscript following your comment. Please, check the attached file and revised manuscript.

Best wishes,

Authors

Reviewer 2 Report

The topic is of great importance and I appreciate the opportunity to review this manuscript. However, for many reasons, the issues are too numerous to be acceptable. The manuscript is listed as a "Design Paper" but it is written as an original research report. Not enough time is spent in the introduction or discussion sections actually describing the study design and the findings of the design itself...

In the introduction, it's mentioned that 25 (line74) and 30 patients (line 78) were recruited.

The formatting of this manuscript also made it hard to follow. Should a clearer form of the last paragraph of the Introduction actually be in the Materials and Methods section?

There are spacing issues in the "Initial evaluation" section of the Materials and Methods.

Why is the first table not in the Results section, when it describes results?

In fact, there are several paragraphs throughout the paper that should be in other sections than the ones they are currently in, adding to my confusion.

The abstract mentions that the nurses were evaluated? Is that included in the manuscript?

Author Response

(The authors gave the same response as above.)

Round 2

Reviewer 1 Report

The authors addressed all concerns that I had regarding this manuscript. I happily recommend this article for publication in International Journal of Environmental Research and Public Health.